# INRAS: Implicit Neural Representation for Audio Scenes

**Kun Su** [*†]         **Mingfei Chen** [*†]         **Eli Shlizerman** [‡†§]

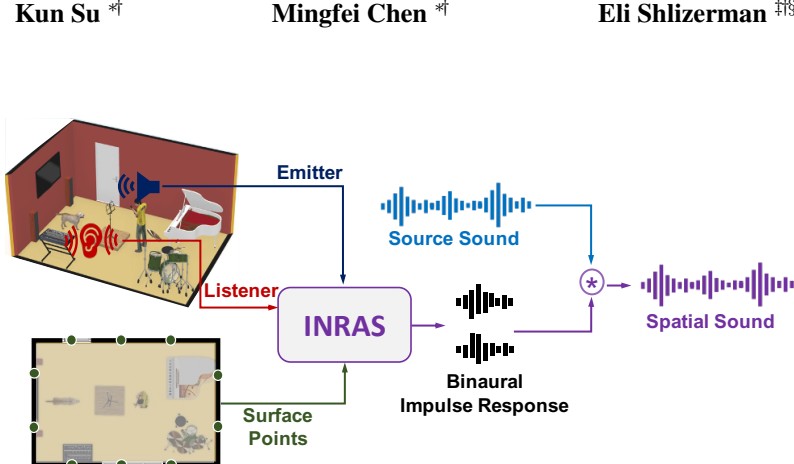

Figure 1: INRAS learns an implicit neural representation for audio scenes such that given the geometry of a scene, emitter and listener positions, INRAS renders the sound perceived by the listener. See supplementary video of demonstration examples of spatial sound rendering.

## Abstract

The spatial acoustic information of a scene, i.e., how sounds emitted from a particular location in the scene are perceived in another location, is key for immersive scene modeling. Robust representation of scene's acoustics can be formulated through a continuous field formulation along with impulse responses varied by emitter-listener locations. The impulse responses are then used to render sounds perceived by the listener. While such representation is advantageous, parameterization of impulse responses for generic scenes presents itself as a challenge. Indeed, traditional pre-computation methods have only implemented parameterization at discrete probe points and require large storage, while other existing methods such as geometry-based sound simulations still suffer from inability to simulate all wave-based sound effects. In this work, we introduce a novel neural network for light-weight Implicit Neural Representation for Audio Scenes (INRAS), which can render a high fidelity time-domain impulse responses at any arbitrary emitter-listener positions by learning a continuous implicit function. INRAS disentangles scene's geometry features with three modules to generate independent features for the emitter, the geometry of the scene, and the listener respectively. These lead to an efficient reuse of scene-dependent features and support effective multi-condition training for multiple scenes. Our experimental results show that INRAS outperforms existing approaches for representation and rendering of sounds for varying emitter-listener locations in all aspects, including the impulse response quality, inference speed, and storage requirements.

---

[*]These authors contributed equally.

[†]Department of Electrical & Computer Engineering, University of Washington, Seattle, USA.

[‡]Department of Applied Mathematics, University of Washington, Seattle, USA

[§]Corresponding author: shlizee@uw.edu

36th Conference on Neural Information Processing Systems (NeurIPS 2022).

# 1  Introduction

There are more than a billion buildings in the world, each of them with unique architecture, interior design and activities they are intended for. While vision is the primary sense for overall impression and navigation through the world's interior scenes, hearing plays a key role for a full immersion in a scene. Indeed, many of our daily activities in an interior scene, such as having a conversation with someone somewhere in the scene, listening to music or watching TV, calling our pets and locating them, are dependent on the hearing function. Hearing is the sense that allows us to experience the scene and interact with it, and the sound quality and its synchronization with the scene completes our audio-visual perception. Indeed, the selection of scene acoustics plays a significant role in the activities that the scene would be used for. For example, a dedicated IMAX theater with the latest sound surround system will draw audience to watch the latest movies, while educational activities will be held in quiet classrooms, and coffee shops with their energetic, but not noisy, environment will draw visitors to work on their laptops. In these examples, spatial sound perception is affected by the collection of the reflected sounds bounced off the floor, the walls, the ceiling, and other reflective surfaces in the scene.

It is thus imperative to computationally model spatial audio aspects of interior scenes in order to adequately render a scene with spatial audio. Computational modeling and representation of spatial audio in an arbitrary scene is a non-trivial task, however, it has been an ongoing research theme with long history in acoustics research [1]. Typically, the relationship between an arbitrary emitter sound and spatial sound can be represented by an impulse response, which is the function of time and positions of the emitter and the listener [2]. For a real scene, an impulse response between the emitter and the listener can be usually measured by playing a sine sweep, using a loudspeaker at the emitter location and recording the sound pressure with a microphone at the listener location [3]. Alternatively, the impulse responses can be also simulated by computational geometry-based sound propagation techniques for scenes that are real or virtual [4, 5, 6, 7]. In both cases, it is time-consuming and computationally expensive to render impulse responses in a continuous space, and therefore prohibit more immersive, interactive spatial sound rendering in scenes. Classic encoding approaches parameterize the impulse responses using a few perceptual parameters that guide reproduction of reverberations [8, 9]. However, such features are typically custom and designed specifically for some scenes and do not guarantee high fidelity impulse responses for a given novel scene.

In this work, we propose an Implicit Neural Representation for Audio Scenes, INRAS, for efficient representation of spatial audio fields with high fidelity. In recent years, neural networks have been shown to parameterize implicit, continuous representations and achieved remarkable progress in computer graphics [10]. The infinite resolution property of such representation could be advantageous for representing the acoustic field as well. Since the acoustic wave equation governs the sound propagation from an emitter in a scene and its solution can be considered as a continuous field of impulse responses, the acoustic field can be encoded via a smooth, continuous representation which can alleviate the drawbacks of the approaches that encode the impulse response in discrete positions and perform interpolation during rendering [8, 9]. Furthermore, our approach is motivated by interactive sound propagation techniques using precomputed acoustic transfer operator for the scene, where the transfer operator is dependent on the scene geometry and decoupled from the emitter and the listener positions to render impulse response efficiently in interactive sound rendering applications [11, 12].

Specifically, INRAS is a light-weight and efficient neural network model that can produce high fidelity spatial impulse responses at arbitrary emitter-listener positions. INRAS includes two main stages. In the first stage, it decomposes the features of the audio scene into three parallel modules: $i$) the Scatter module, $ii$) the Bounce module, and $iii$) the Gather module. Motivated by the disentangled procedures in the interactive acoustic radiance transfer techniques [11, 12], we design these three modules to generate independent features for the emitter, scene geometry, and listener, respectively. Indeed, disentangling the features of scene geometry allows our model to represent multiple scenes by adding only a few trainable parameters. In the second stage, the listener module fuses the independent features and generates the directional and binaural impulse responses. We show an overview of INRAS in Figs 1 and 3.

In summary, our main contributions in this work are: 1) We propose a novel light-weight neural network, INRAS, to learn an implicit neural representation for audio scenes that generate high fidelity time-domain impulse responses for arbitrary emitter-listener. 2) INRAS disentangles scene's

geometry features to generate independent features for the emitter, the geometry of the scene, and the listener, which leads to efficient reuse of scene-dependent features for arbitrary emitter-listener positions, and supports effective multi-condition training for multiple scenes by adding only a few trainable parameters. 3) INRAS outperforms existing approaches on all metrics of audio rendering, including the impulse response quality, inference speed, and storage requirements.

## 2 Related Work

***Scene Acoustics Modeling.*** Modeling scene acoustics can be divided into two categories, approaches of 1) wave-based and, 2) geometry-based. The first type of wave-based algorithms aims to solve the acoustic wave equation using numerical techniques [13, 14, 15, 16]. Due to the computation complexity of the wave equation, these approaches are typically used for lower frequencies. While wave methods have become more utilized with advancement of CPU/GPU computing power [17, 18], this cost directs existing methods to prefer geometric approximations of scene acoustics (the second type of acoustic modeling) [19]. Geometry-based approaches assume that the sound travels along a straight line, and determine the path of sound propagation according to energy attenuation. These methods are generally faster than wave-based methods and are suitable for high-frequency sound propagation. However, with such an approach, it is difficult to accurately simulate low-frequency acoustic phenomena such as edge diffractions and surface scattering of arbitrary order. The commonly used geometric approaches are image sources methods [4, 5], ray-tracing [6, 7, 20], radiosity [21], and acoustic radiance transfer [22].

A general model of geometric room acoustics can be formulated as an integral equation. One of the first equations is the Kuttruff's integral equation for diffuse reflections in a convex room [23]. Multiple extensions of this mathematical model have been proposed subsequently, such as the room acoustic rendering equation which provides a framework for most geometric acoustic methods for interiors [24]. These algorithms for sound propagation are limited to static sources and/or listeners. Interactive applications are usually achieved by precomputing sound propagation effects such as precomputing acoustic radiance transfer from static sources [11, 12, 25]. While our work aims to represent the scene acoustics instead of performing simulation from scratch, the proposed INRAS model is motivated by the interactive acoustic radiance transfer method [11, 12].

***Sound Field Encoding.*** Classical sound field encoding approaches represent the field around a listener point by capturing the sound from spatially distributed sources. For example, Ambisonics [26] represents the sound field around a point using spherical harmonic coefficients and independently of the reproduction setup (speakers or headphones). Parametric surround approaches, such as MPEG-Surround [27], assume a known speaker configuration around the listener. MPEG-H [28] extends the idea to allow encoding that is agnostic to the reproduction setup and supports higher-order Ambisonics and binaural rendering. The Spatial Decomposition Method (SDM) [29] fits an image source model to responses measured with a microphone array, approximating it at a point with multiple delayed spherical wavefronts. In Directional Audio Coding (DiRaC) [30], the input is the directional sound signal at a listener which is a superposition of all sound source signals in a scene convolved with the corresponding directional impulse responses. DiRaC computes direction and the diffuseness parameter for each of many time-frequency bins. These approaches are static and do not allow the listener to navigate the scene and experience the changes in sound while doing so. Several works for interactive sound field encoding propose to extract important features from precomputed impulse responses and synthesize them back using digital signal processing techniques [8, 31, 9]. However, these encodings typically cannot reproduce impulse responses with high fidelity.

***Deep Acoustics.*** In recent years, deep learning approaches have been developed for various audio applications, especially in speech [32, 33, 34, 35, 36] and music [37, 38, 39, 40]. In acoustics, these include neural sound spatialization from a mono audio [41], estimation of room geometry and reflection coefficients from impulse response [42], reverberation time and direct-to-reverberation ratio prediction [43, 44], and learning the head-related transfer functions (HRTFs) [45]. In relation to scene modeling, deep neural networks which model room impulse responses (RIR) have been extensively studied. A convolutional neural network model has been proposed to estimate room impulse response from reverberant speech [46]. Deep generative models such as IR-GAN [47] and fast-RIR [48] have been proposed to generate new realistic impulse responses. Recently, the emergence of implicit neural representations has shown great success in representing 3D geometry [49] and the appearance of a scene [10]. Such representation approach could be generalized to represent images, videos, and

sounds by learning a continuous mapping capable of capturing data at an "infinite resolution" [50]. Indeed, very recently, it has been proposed to learn an implicit neural function to represent the impulse responses of interior scenes such as rooms [51, 52]. The Impulse Response Multi-layer perceptrons (IR-MLP) proposes to predict impulse responses from spatio-temporal coordinates using an MLP. However, this approach does not support both moving sources and moving listeners scenarios [51]. Such a problem has been addressed by Neural Acoustic Fields (NAF) [52], which proposes to learn a continuous map from all emitter and listener location pairs to a neural impulse response function using the magnitude component of the frequency-time spectrogram representation after applying Short-time-Fourier-Transform. While the smooth nature of the time-frequency spectrogram can be beneficial for training deep neural networks, the smoothness and entanglement of the time-frequency representation prediction also leads to imprecise modeling of high peaks that appear less frequently. For example, the sparse high peaks in the early reflection part of impulse response play a dominant role in our perceptual feelings for sound source directions and clarity. Moreover, modeling using the spectrogram magnitude ignores the phase information and adding a random phase which may distort the audio signal. In our approach, we learn the neural representation of the sound field for both *moving listeners* and *moving sources* scenarios. We aim to learn such implicit neural representation for rendering time-domain impulse responses instead of spectrograms. Our results show that INRAS can generate higher fidelity impulse responses with even fewer trainable parameters.

## 3   Methods

***Problem Setup.*** INRAS implements several deep neural networks to model the continuous implicit function that maps the coordinates of the scene to the corresponding time-domain directional and binaural impulse responses of the sound field. More formally, for a given 3D scene $D$, we denote the sound emitter locations as $s \in \mathbb{R}^3$, the listener locations as $l \in \mathbb{R}^3$, and the listener head orientation $\theta \in \mathbb{R}^2$. Then $\forall (s, l, \theta) \in \mathbb{R}^8$ in the scene, there would be corresponding binaural impulse responses $h \in \mathbb{R}^{2 \times T}$ where $T$ indicates the time length. We model the continuous function $f(s, l, \theta) \rightarrow h$ parameterized by a deep neural network that pairs $s, l, \theta$ with appropriate impulse response $h$. While the idea seems straightforward, training the network to learn the time domain impulse response from given coordinate inputs is challenging due to typical long temporal length of impulse responses, and highly oscillating amplitude at different time samples, all which increase the difficulty of training. A key insight to assist with training is that while the geometry of the scene determines the impulse responses, it is always static no matter how emitter and listener positions vary and therefore the geometry based information could be shared with an arbitrary emitter and listener positions. Such an idea has been applied to interactive sound propagation based on acoustic radiance transfer [11, 12]. For training a neural network model, the approach would be to leverage the static scene geometry by learning reusable scene-dependent features, and associate them with the emitter and the listener. This allows the model to realize that the differences between impulse responses at various emitter-listener locations are dependent on the scene geometry. Motivated by this approach, we propose a two-stage model. The first stage performs decomposition of the audio scenes feature to learn the independent scene geometric features and associate the emitter and listener to the scene. The second stage fuses these features to render the binaural impulse responses. In the following sections, we review the interactive acoustic radiance transfer and then describe our model in detail.

***Background on Interactive Acoustic Radiance Transfer.*** The acoustic radiance transfer is a classical approach to model sound propagation in complex room models and it can be derived from the acoustic rendering equation [24]

$$L(x, \Omega, t) = L_0(x, \Omega, t) + \int_S R(x, x', \Omega, t) L(x', \frac{x - x'}{|x - x'|}, t) dx',  \tag{1}$$

where $S$ is the set of all surface points in the scene, $L$ is the total outgoing acoustic radiance, $L_0$ is the emitted acoustic radiance, $\Omega$ is the final radiance direction at $x$; the incident radiance direction at $x$ is implicit in the specification of $x'$, and $R$ is the reflection kernel, which describes how radiance at point $x'$ influences radiance at point $x$. The equation describes that the outgoing time-dependent radiance at any surface point is a combination of the reflected time-dependent radiance and the emitted time-dependent radiance.

The acoustic radiance transfer algorithm can be summarized in three steps (See Fig. 2). In the first step, the scene's boundary is divided into $N$ bounce points, and energy is scattered from the emitter to all bounce points. In the second step, sound energy is emitted in all directions from a given bounce point. It propagates through the scene until the propagation is finally terminated upon an

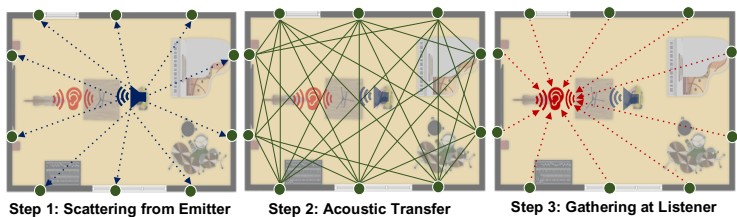



**Step 1: Scattering from Emitter**    **Step 2: Acoustic Transfer**    **Step 3: Gathering at Listener**

Figure 2: Acoustic radiance transfer steps overview.



incidence at some other bounce point. The energy-time curve on each bounce point can be stored as an echogram. In the final step, the listener gathers energy responses from all bounce points. In interactive extensions [11, 12], a linear acoustic transfer operator is precomputed to model the propagation of acoustic radiance between bounce points distributed over the surface of the scene. In other words, the acoustic transfer operator can be seen as the scene-dependent features that are shared with all emitter-listener locations. Such disentanglement efficiently updates the impulse response at various emitter-listener positions by computing the propagation delay based on the relative distance to the bounce points. This motivates us to design a neural network model with similar decoupled modules to satisfy that the scene geometry information can be realized and reused by an arbitrary emitter and listener.

***Implicit Neural Representation for Audio Scenes.*** INRAS includes two main components:$(a)$ audio scenes feature decomposition, and $(b)$ spatial binaural impulse response prediction. In $(a)$, there are three parallel modules: 1) the Scatter module which learns features to associate the emitter with bounce points; 2) the Bounce module which learns the scene-dependent features shared by all emitter and listener positions; 3) the Gather module which learns features to associate the listener with the bounce points. In $(b)$, we fuse the output features of the three parallel modules and render the directional and binaural impulse responses. System overview is shown in Fig. 3.

*Scatter Module.* Similar to computing the initial radiance scattering from the emitter to all bounce points in acoustic radiance transfer, the Scatter module is dependent on the relative distance between the emitter position and every bounce point position. We divide the surface of the scene into $N$ uniform bounce points with 3D locations $\{b_i\}_{i=1}^N \in \mathbb{R}^3$. We compute the relative distance between the emitter position $s$ to all bounce points $\{d_{b_i}^s\}_{i=1}^N$. Using relative distance as input instead of absolute position enables the emitter to be aware of the geometry of the scene and allows the model to learn smooth continuous features for various emitter positions. We use sinusoidal encoding to map the input $\{d_{b_i}^s\}_{i=1}^N$ to a higher dimension. This encoding is similar to the encoding used in graphical implicit neural representation [10]. We learn a function $F_\Theta$ parameterized by a fully connected network. We denote the output feature as $I = F_\Theta(\{d_{b_i}^s\}_{i=1}^N) \in \mathbb{R}^{N \times D}$, where $D$ indicates the feature dimension. In our experiments, we find that it is sufficient to use 40 to 60 bounce points to cover the structure of the scene. We perform more investigations of bounce points selection in ablation studies Results section.

*Bounce Module.* We design the bounce module to generate features representing the geometry of static scenes shared with arbitrary locations of emitter and listener. To model such scene dependent features, we learn a function $U_\Phi$ parameterized by a multi-layer perceptron (MLP) with residual connections that takes all bounce points positions $\{b_i\}_{i=1}^N \in \mathbb{R}^3$ as input and outputs the features $Q = U_\Phi(\{b_i\}_{i=1}^N) \in \mathbb{R}^{N \times D}$.

*Gather Module.* This module is similar to the Scatter module. We aim to associate the listener with the bounce points in the scene. We compute the relative distance between the listener position $l$ to all bounce points: $\{d_{b_i}^l\}_{i=1}^N$, and use sinusoidal encoding to learn a function $G_\Psi$ parameterized by a fully connected network to generate the output feature $O = G_\Psi(\{d_{b_i}^l\}_{i=1}^N) \in \mathbb{R}^{N \times D}$.

*Spatial-Time Feature Composition.* The three modules in Feature Decomposition described above do not incorporate time-dependencies. Adding time to every module could significantly slow down the training procedure. Motivated by the acoustic operator decomposition in the interactive sound propagation [12], the energy-time echogram for a specific bounce point $b_i(t)$ can be represented by a set of time domain basis functions $\{\tau^k(t)\}_{k=1}^K$ via a linear combination: $b_i(t) = \sum_{k=1}^K \alpha_k \tau^k(t)$, where $\alpha$'s are coefficients in the basis space. We follow this representation and learn a function

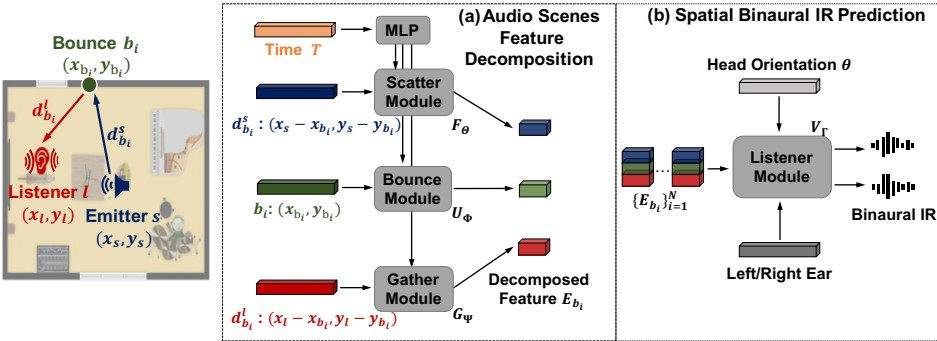

Figure 3: System overview of INRAS. (a) Audio Scenes Feature Decomposition: inputs to scatter/gather module are the relative distances between the emitter/listener locations and bounce points. The bounce module takes all bounce points to generate scene-dependent features. (b) Spatial Binaural IR Prediction: in this stage, the decomposed features are stacked and fed to the Listener module which generates the spatial binaural impulse responses.

$P_\tau$ through a fully connected network to obtain a set of time-domain basis functions which can be reused by all spatial features. We encode the time samples $\{t_j\}_{j=1}^T$ using the sinusoidal encoding. The output is denoted as $M = P_\tau(\{t_j\}_{j=1}^T) \in \mathbb{R}^{T \times D}$. We then perform fast matrix multiplication to obtain spatial-time features $\hat{I} = MI^\top$, $\hat{Q} = MQ^\top$ and $\hat{O} = MO^\top$.

*Listener Module.* In the stage of spatial binaural impulse response prediction, the listener module first performs feature fusions by concatenating the three features together $E = \{\hat{I}, \hat{Q}, \hat{O}\} \in \mathbb{R}^{T \times 3N}$, where $\{E_{b_i}\}_{i=1}^N \in \mathbb{R}^{T \times 3}$ represents fused spatial-time features for $l$ and $s$ associated with the bounce point $b_i$. We feed $E$ as input to the Listener module which combines it with the orientation of the head given by $\theta$ and encoded by a learnable embedding matrix. We model the listener module via MLP and generate binaural impulse responses in time-domain $h = V_\Gamma(E, \theta)$.

***Training and Rendering.*** All components and modules of INRAS are trained jointly. We use a combination of mean square error loss $L_{\text{mse}} = \|h - \hat{h}\|_2^2$ and multi-resolution STFT loss $L_{\text{mr\_stft}}$ which has been shown effective in modeling audio signals in the time domain [53]. The multi-resolution STFT loss first converts the impulse response into frequency-time domain $H = \text{STFT}(h)$ and computes the spectral convergence loss $L_{\text{sc}} = \frac{\||H| - |\hat{H}|\|_2}{\||H|\|_2}$, the magnitude loss $L_{\text{mag}} = \||H| - |\hat{H}|\|_1$ and the phase loss $L_{\text{phase}} = \|\phi(H) - \phi(\hat{H})\|$, our total loss can be summarized as follow:

$$L_{\text{mr\_stft}} = L_{\text{sc}} + L_{\text{mag}} + L_{\text{phase}}, L_{\text{total\_loss}} = L_{\text{mse}} + L_{\text{mr\_stft}} \tag{2}$$

Once we obtain the impulse response $h$, we can render sounds perceived at the listener location by convolving the impulse response with a sound source $y$. The final sound is denoted as $\hat{y} = h \circledast y$.

***Multi-condition Training with Multiple Scenes.*** The design of INRAS enables the emitter and the listener to be aware of scene geometry by computing the relative distance to the bounce points in Scatter and Gather modules and provides a static scene-dependent features. This concept could be generalized to multiple scenes, since one can include bounce points from multiple scenes and let the emitter and listener to predict which scene they are in to achieve a multi-condition representation goal. Specifically, we normalize the coordinate space of multiple scenes and adapt the total number of bounce points $N_{\text{total}} = \sum_{i=1}^K N_i$ for $K$ scenes. When computing the relative distance and bounce points features for the emitter/listener in a specific scene, we mask the other irrelevant bounce points. Since all other components and feature dimension are kept the same, such operation adds a handful of trainable parameters due to the increased bounce points number and in turn enables the multi-condition representation over multiple scenes.

## 4 Experiments

***Datasets.*** To evaluate our method, we use the *Soundspaces* dataset which consists of dense pairs of impulse responses generated by geometric sound propagation methods [54]. All scenes have

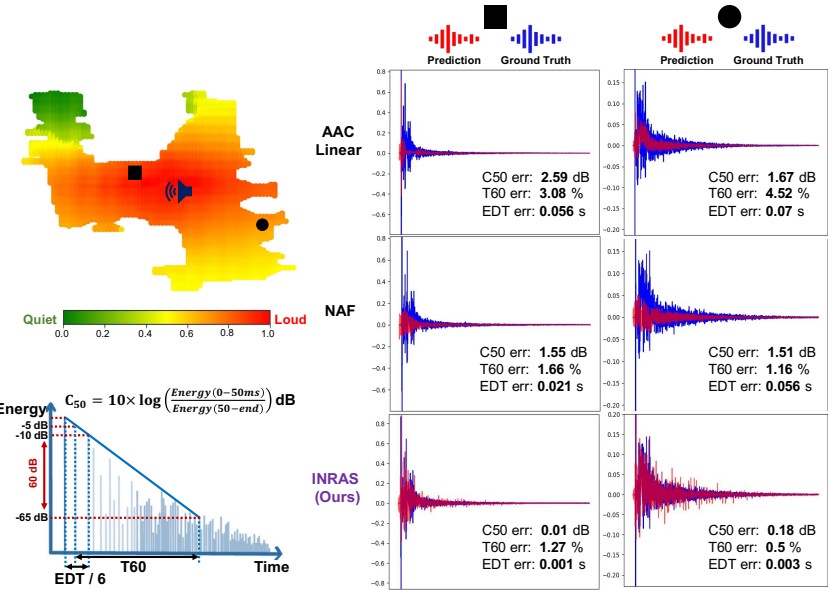

Figure 4: Rendered Impulse Responses Waveform Visualization. Top left:the speaker indicates the emitter location. We show examples (right) of rendered waveforms at two listener locations (black square and black circle) rendering by three methods: AAC-Linear, NAF and INRAS (blue: GT; red: Prediction). Bottom left: Metrics upon which we evaluate Impulse Response are illustrated.

the same ceiling height and provide binaural impulse responses for four different head orientations $(0, 90, 180, 270$ degrees). For a fair comparison to previous work [52], we re-sample all impulses responses to 22050 sampling rate and use the same 6 scenes, including 2 multi-room layouts, 2 rooms with non-rectangular walls, and 2 single rooms with rectangular walls. For each scene, we use 90% data for training and hold 10% data for testing.

***Implementation Details.*** We use Pytorch to implement all INRAS models. For all scenes, we extract the bounce points from the mesh boundary, (40 to 60, depending on the scene). We encode the relative distance from emitter/listener to bounce points using sinusoidal encoding with 10 frequencies of sin and cos functions. We use a fully connected layer in the Scatter module and Gather module. In the Bounce module, we use a 4-layer residual MLP. In the Listener module, we use a 6-layer residual MLP. In all MLPs, we use 256 neurons and set PreLU as the activation function. We use AdamW optimizer [55] to train all models on a Tesla T4 GPU for 100 epochs with a batch size of 64. The initial learning rate is set as 5e-4 and it gradually decreases by a factor of 0.95.

***Baseline Methods.*** We compare our method to existing learning-based and classical approaches. For learning-based approaches, we compare INRAS with NAF [52]. Since INRAS is effectively a *novel encoding* method to compress the acoustic field of a scene, we also compare INRAS with two *audio encoding* methods: Advanced Audio Coding (AAC) and Xiph Opus by applying both linear and nearest neighbor interpolation to the coded acoustic fields.

***Evaluation Metrics.*** We evaluate all methods on three aspects: impulse response quality, storage requirements and inference speed. We first compute acoustic parameters to evaluate the impulse response quality. We use acoustic parameter Clarity (C50) to quantify the part of early reflections of the impulse response which is associated with music loudness, speech intelligibility, and clarity. To study the effects of late reverberation parts, we use reverberation (T60) and early decay time (EDT) to illustrate the statistical portion of the impulse response. The reverberation time (T60) measures how long it takes for the acoustic energy to decay by 60 dB. EDT is closely related to the listener's perception of reverberation but it is also affected by the early reflections of the impulse responses. Illustration for these acoustic metrics can be found in Fig. 4. In addition, we also compute the storage requirements for saving audio scenes representations and the inference speed for rendering a binaural impulse response in the scene. For fair comparison, we test inference speed for all methods consistently on a same Telsa T4 GPU.

| Model\Metric | C50 error (dB) ↓ | T60 error (%) ↓ | EDT error (sec) ↓ | Parameters (Million) ↓ | Storage (MB) ↓ | Speed (ms) ↓ |
|---|---|---|---|---|---|---|
| Opus-nearest | 3.58 | 10.10 | 0.115 | - | 181.37 | - |
| Opus-linear | 3.13 | 8.64 | 0.097 | - | 181.37 | - |
| AAC-nearest | 1.67 | 9.35 | 0.059 | - | 346.74 | - |
| AAC-linear | 1.68 | 7.88 | 0.057 | - | 346.74 | - |
| NAF | 1.06 | 3.18 | 0.031 | 2.23 | 8.55 | 37.86 |
| **INRAS (Ours)** | **0.6** | **3.14** | **0.019** | **0.67** | **2.56** | **9.47** |

Table 1: Quantitative evaluation of impulse response quality, storage requirements and inference speed. Results are indicated on average of six single scene models.

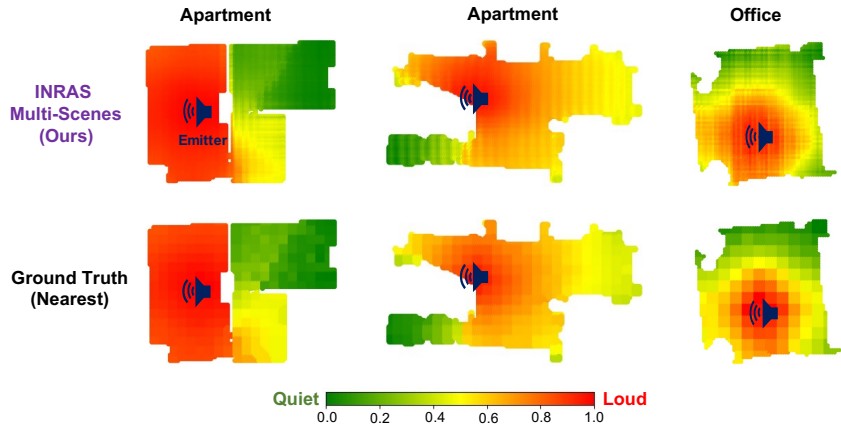

Figure 5: Loudness map visualization comparing INRAS multi-scenes rendering of three scenes (Top) vs. Ground truth using nearest neighbors (Bottom).

***Results.*** Quantitative evaluation results are shown in Table 1. INRAS outperforms both traditional audio coding and learning-based methods in all metrics. In particular, C50 and EDT errors for INRAS outperform NAF by $43\%$ and $39\%$, indicating that the early reflection part of INRAS rendered impulse responses is much closer to the ground truth. Fig. 4 illustrates a comparison of two examples of rendered impulse responses waveforms by AAC-linear, NAF and INRAS method. On the top left of the figure, we visualize the impulse responses loudness map of INRAS where colors indicate the loudness amplitude. In the two right columns, the comparison shows that the AAC-linear waveforms have large gaps from the ground truth. While NAF is able to capture the exponential decay pattern of reverberation, it cannot capture the early reflection part of both impulse responses that are shown. This part includes high peaks important for clarity. In comparison, INRAS can render both the early reflections and the late reverberation. We show more qualitative visualization on loudness maps and waveforms in the Suppl. Materials. Moreover, INRAS model takes only $\sim 0.65$ million trainable parameters which results in less than 3MB storage and 4ms inference speed, indicating the INRAS is significantly more light-weight and efficient model.

***Multi-condition Training on Multiple Scenes.*** As discussed in the method section, the effective audio scene feature decomposition allows us to train a single INRAS model to represent multiple scenes. We investigated this property by training INRAS on three scenes with different types of scene layouts. We selected one multi-room layout, one room with non-rectangular walls, and one room with rectangular walls (See Fig. 5). As expected, INRAS can learn continuous implicit neural representations for all three scenes. We illustrate the loudness maps for all three scenes learned by such single model in Fig. 5 and in Table 2, where we show quantitative results of the multi-scene model. In addition to the acoustic parameters that evaluate the impulse response quality, we further evaluate the quality of the final rendered audio signal after convolving the impulse response with a sound source. Specifically, we compute the Signal-to-Noise ratio (SNR) and audio Peak Signal to Noise Ratio (PSNR). The results in Table 2 clearly show that our model after multi-condition training can achieve high-quality results and better overall accuracy than average of other models applied to each scene. Notably, the number of trainable parameters in INRAS increases only by $0.1M$ to extend the single-scene to multi-scenes thus keeping the storage requirement less than 3MB. In comparison, other approaches have increased the storage size linearly.

| Model\Metric | Multi-scenes | SNR (dB) ↑ | PSNR (dB) ↑ | C50 error (dB) ↓ | T60 error (%) ↓ | Storage (MB) ↓ |
|---|---|---|---|---|---|---|
| Opus-nearest | ✗ | 3.18 | 13.35 | 3.6 | 10.1 | 544.11 |
| Opus-linear | ✗ | 3.57 | 13.45 | 3.23 | 8.7 | 544.11 |
| AAC-nearest | ✗ | 6.48 | 17.84 | 1.51 | 9.64 | 1040.31 |
| AAC-linear | ✗ | 7.52 | 18.7 | 1.57 | 8.05 | 1040.31 |
| NAF | ✗ | -1.54 | 11.25 | 1.05 | **3.01** | 25.65 |
| **INRAS (Ours)** | ✓ | **8.06** | **18.80** | **0.68** | 4.09 | **2.99** |

Table 2: Quantitative evaluation of INRAS after multi-condition training on three scene layouts. Results for other methods are computed as an average of the three scenes.

*Ablation Studies.* To show the effectiveness of INRAS v.s. similar variants, we use a representative scene to perform ablation studies. Table 3 shows comparison results of INRAS and its ablated variants. We first implement a brute-force model (Simple INRAS) using a residual MLP like NAF architecture and provide the normalized emitter and listener positions as input to predict the time domain impulse response using MSE loss only. The result turns out to be unsuccessful in all metrics. We further show that adding a multi-resolution STFT loss can improve T60 error but still the model fails to capture the early reflection part. Next, we show that not including the relative distance impairs the results since the emitter and the listener could not realize the scene geometry. Also, removing the bounce module eliminates the static scene feature and therefore impairs the performance. We also investigate the importance of bounce point selection. We sample two types of bounce points that both have the same total number as the original setting but they do not cover the whole scene, i.e., missing some boundaries. The results show that only using bounce points cover that the full scene geometry can achieve the best performance in all metrics.

| Model\Metric | C50 err (dB) ↓ | T60 err (%) ↓ | EDT err (sec) ↓ |
|---|---|---|---|
| Simple INRAS w. $L_{mse}$ | 1.47 | 49.6 | 0.048 |
| Simple INRAS w. $L_{mse} + L_{mr\_stft}$ | 2.20 | 6.40 | 0.074 |
| INRAS w.o. rel. dist. | 1.12 | 3.52 | 0.038 |
| INRAS w.o. bounce module | 0.63 | 2.30 | 0.019 |
| INRAS w. more incomplete bounce points | 0.50 | 2.31 | 0.019 |
| INRAS w. less incomplete bounce points | 0.49 | 2.17 | 0.018 |
| **INRAS (Ours)** | **0.44** | **2.07** | **0.017** |

Table 3: Ablation Studies of INRAS variants.

*Bounce Points Sampling.* We uniformly sample bounce points on the boundaries of obstacles appearing in the scene. In the Soundspace dataset, the only obstacles are walls that block sound propagation. Thus we uniformly sample the points on the surface of each room. The number of bounce points is dependent on the room size and the sampling rate. We performed experiments with various sampling rates to evaluate which selection to use. For example, one room has a perimeter of about 24 meters and the total number points of the mesh boundary is around 2400. Our experiments indicated that uniform sampling of the bounce points every 0.5m (sample rate: $1/50$, 48 bounce points) is optimal. Notably, the impulse response data has the same spatial resolution. A higher sampling rate does not necessarily achieve better performance since noise and redundant features could be induced with higher sampling resolution. We compare different sampling rates for bounce points for the room mesh in Table 4.

*Multi-conditioned Representation for Various Reflection Coefficients.* To test whether INRAS can be conditioned on various reflection coefficients, we create a 3m x 5m room using the available open source Python package (pyroomacoustics) [56] to simulate impulse responses using Image Source + Ray tracing method. We use spatial resolution of 0.5m to sample both impulse responses and bounce points. For the same room, we create 225 sets of impulse responses with various absorption and

| Sampling Rate\Metric | T60 error (%) ↓ | C50 error (dB) ↓ | EDT error (sec) ↓ |
|---|---|---|---|
| 1/100 | 1.47 | 0.53 | 0.019 |
| 1/75 | 1.42 | 0.53 | 0.018 |
| **1/50** | **1.41** | **0.51** | **0.016** |
| 1/25 | 1.41 | 0.59 | 0.019 |
| 1/10 | 1.70 | 0.71 | 0.025 |

Table 4: Sampling rate study for bounce points sampling.

| Settings\Metric | T60 error (%) ↓ | C50 error (dB) ↓ | EDT error (sec) ↓ |
|---|---|---|---|
| Seen Coefficients | 3.56 | 1.06 | 0.023 |
| Unseen Coefficients | 3.83 | 1.13 | 0.030 |

Table 5: Results on multi-conditioned reflection coefficients. These results show that INRAS can perform well even when absorption and scattering coefficients are unseen and vary.

| Method\Metric | T60 error (%) ↓ | C50 error (dB) ↓ | EDT error (sec) ↓ |
|---|---|---|---|
| Geometry Based Simulation | 1.45 | 3.67 | 0.042 |
| INRAS (ours) | 0.35 (-75.86%) | 1.61 (-56.13%) | 0.011(-73.81%) |

Table 6: Comparison between INRAS (ours) and Geometry Based Simulation for a real world scene. Our results show that INRAS outperforms geometry-based simulation in all metrics and obtains significant improvement in accuracy.

scattering coefficients on the walls. We use $90\%$ sets of coefficients for training and $10\%$ for testing. For the coefficients seen in the training, we also hold $10\%$ spatial points for testing. To incorporate the coefficient condition in our model, we adapt our model by adding a coefficient embedding using sinusoidal embedding similar to coordinate inputs. An addition operation is performed to add the coefficient embedding to the decomposed features. In this experiment, as shown in Table 5, INRAS model achieves testing T60 error of $3.83\%$, C50 error of $1.13$dB and EDT error of $0.03$ second, which is not far from the performance on the training set. These results show that INRAS can perform well even when absorption and scattering coefficients are unseen and vary.

## 5   Conclusion

In conclusion, here we present INRAS, a novel implicit neural representation for audio scenes. INRAS is a light-weight, fast model that effectively renders high fidelity impulse responses for multiple audio scenes. We achieve such function by leveraging a novel reusable representation of scene-dependent features and associate them with the emitter and the listener. Experimental results demonstrate that INRAS outperforms other methods in all metrics and we further show that INRAS can represent multiple scenes after multi-condition training.

There are multiple benefits of an implicit neural representation of acoustic fields by INRAS. The first benefit is that in comparison to previous pre-computation approaches, an implicit neural representation of INRAS can learn a continuous compact representation for pre-computed impulse responses in a scene. This significantly decreases the storage requirement and provides fast inference in run-time. Some state-of-the-art geometry-based sound simulations can be made real-time as well. However, these approaches still have the common drawbacks of geometric-based methods, such as inaccuracy at low frequency and inability to simulate complete wave-based sound effects. In comparison, an implicit neural representation of INRAS is not limited to these drawbacks at run-time since conceptually it can learn to map input coordinates to infinite-high fidelity impulse responses which can be pre-computed. In other words, the complexity of the sound simulation will not be a bottleneck for the quality of the real-time rendering.

Furthermore, implicit neural representations can fit real world data, on which it is challenging for an interactive geometric-based simulation to reach the same quality. To further investigate this aspect, we trained INRAS on S1-M3969 subset of meshRIR dataset which includes dense real impulse responses recordings in a room with approximate dimensions of 7m x 6.4m [57]. We sample the bounce points every $0.5$m. Since the recording is a single channel, we removed the head orientation conditions at the listener module and output a single channel. We use $90\%$ data for training and $10\%$ data for testing. For a fair comparison, we faithfully tuned material parameters of the room model to best match the ground truth recordings. Our results show that INRAS outperforms geometry-based simulation in all metrics and obtains large gains in accuracy, as shown in Table 6.

**Acknowledgement** We acknowledge the support of National Science Foundation grant OAC-2117997.

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
