# OpenReview forum: "INRAS: Implicit Neural Representation for Audio Scenes"
_NeurIPS.cc/2022/Conference — NeurIPS 2022 Accept_

### Official Review · Reviewer_RoFe · 2022-07-11

**Rating:** 5
**Confidence:** 4
**Soundness:** 3 good
**Presentation:** 2 fair
**Contribution:** 2 fair

**Summary:**

This paper proposes to use implicit neural representation to represent spatial audio in the scene. It renders high fidelity time-domain impulse responses at any arbitrary emitter-listener positions using neural network parameterization. Experimental results demonstrate that INRAS outperforms existing methods across a series of metrics.

**Questions:**

1. It is claimed "the proposed method renders high fidelity time-domain impulse responses at any arbitrary emitter-listener positions using neural network parameterization", but SoundSpaces only provides impulse responses at a discrete set of grid points in the spatial environment. How the evaluation is done at the points where there is no ground truth?

2. See weakness.

3. How sensitive is the performance to the number of bounce points used?

**Limitations:**

Yes, there are discussions of limitations and potential negative societal impact of the work in the supplementary materials.

**Strengths And Weaknesses:**

Strengths:

- The paper is generally well-written and well-motivated. It nicely reviews prior methods of scene acoustic modeling: wave-based vs. geometry-based. The proposed method is motivated by prior methods on interactive acoustic radiance transfer, and proposes a neural network-based approach to mimic their settings.

- Dividing the whole pipeline into three sub-steps: scattering from the emitter, acoustic transfer, and gathering at listener makes sense. The design of INRAS well captures the physical process of how sound is propagated from the emitter to the listener.

- Compared to the baseline method NAF, the proposed method has large gains.

Weaknesses:

- The proposed method is motivated by prior methods on interactive acoustic radiance transfer, which is nice. And it is cool to represent spatial audio using a neural representation. However, what is the use of it and why a neural implicit representation is better is not discussed. The neural network basically tries to model how sound is propagated from the emitter to the listener. Why not just directly doing geometry-based audio simulation, for example bi-directional path tracing, to obtain the room impulse response. What is the motivation to use an implicit neural representation to replace the spatial audio simulation? Bi-directional path tracing can also be made real-time with some simplifications.

---

> ### Author Response · Authors · 2022-08-02
> **Response to Reviewer RoFe**
>
> We thank the reviewer for a thoughtful review and valuable feedback. Below we answer questions, and as a result the revisions that we will include in the camera ready version of the manuscript which address the reviewer's concerns.
> ### Strengths And Weaknesses
> - W1. We summarize the benefits of using an implicit neural representation:
>   - In comparison to previous pre-computation approaches, an implicit neural representation can learn a continuous compact representation for pre-computed impulse responses in a scene, which significantly decreases the storage requirement and provides fast inference in run-time.
>   - Indeed some state-of-the-art geometry-based sound simulations can be made real-time. However, these approaches still have the common drawback of geometric-based methods, such as inaccuracy at low frequency and inability to simulate complete wave-based sound effects. In comparison, an implicit neural representation is not limited to these drawbacks at run-time because conceptually it can learn to map input coordinates to infinite-high fidelity impulse responses which can be pre-computed ahead of time. In other words, the complexity of the sound simulation will not be a bottleneck for the quality of the real-time rendering. Therefore, we believe that using an implicit neural representation is a promising direction for future sound rendering techniques.
>   - Implicit neural representations can fit real world data, on which it is challenging for an interactive geometric-based simulation to reach the same quality. To further investigate this aspect, we train INRAS on S1-M3969 subset of meshRIR dataset [1] which includes dense real impulse responses recordings in a room with approximate dimensions of 7m x 6.4m. We sample the bounce point every 0.5m. Since the recording is a single channel, we remove the head orientation conditions at the listener module and output a single channel. We use 90% data for training and 10% data for testing. \
> To perform the comparison, we created a room with the same dimension using available open source python package (pyroomacoustics) [2] to simulate impulse responses using Image Source + Ray tracing method. For a fair comparison, we faithfully tuned material parameters of the room model to best match the ground truth recordings. Our results show that INRAS outperforms geometry-based simulation in all metrics and obtains large gains on them as shown in the following table:
> | **Real World Scene**      | C50 error      | T60 error      | EDT error       |
> |---------------------------|----------------|----------------|-----------------|
> | Geometry Based Simulation | 1.45           | 3.67           | 0.042           |
> | INRAS (ours)              | 0.35 (-75.86%) | 1.61 (-56.13%) | 0.011 (-73.81%) |
>
> We intend to include this discussion and the experiment in the revised version of the paper.
> ### Questions
> - Q1. For points that do not have ground truth, we refer to the qualitative results shown on the loudness map and impulse response waveforms in Fig 4, 5 and supplementary material, and demo videos. The loudness map shows that INRAS can render a continuous acoustic field at any location and resemble the ground truth (smooth version). The impulse response waveform also shows a high similarity to the ground truth.
> - Q2. See the response to W1.
> - Q3.  The number of bounce points is dependent on the room size and the sampling rate. We performed experiments with various sampling rates to evaluate which selection to use. For example, ‘Room_2’ has a perimeter of about 24 meters and the total number points of the mesh boundary is ~ 2400. Our experiments indicated that uniform sampling of the bounce points every 0.5m (sample rate:1/50, 48 bounce points) is optimal. Notably, the impulse response data has the same spatial resolution.
> Using a higher sampling rate doesn’t achieve better performance because noise and redundant features are induced as the sampling resolution is higher than the spatial resolution. We compare different sampling rates for bounce points for the mesh of ‘Room_2’ in the following table, which will be added to the supplementary information along with the above discussion.
> | Sampling Rate | T60 error | C50 error | EDT error |
> |---------------|-----------|-----------|-----------|
> | 1/100         | 1.47      | 0.53      | 0.019     |
> | 1/75          | 1.42      | 0.53      | 0.018     |
> | **1/50**      | **1.41**  | **0.51**  | **0.016** |
> | 1/25          | 1.41      | 0.59      | 0.019     |
> | 1/10          | 1.70      | 0.71      | 0.025     |
>
> [1] S. Koyama, et al, "MESHRIR: A Dataset of Room Impulse Responses on Meshed Grid Points for Evaluating Sound Field Analysis and Synthesis Methods," 2021 IEEE Workshop on Applications of Signal Processing to Audio and Acoustics (WASPAA), 2021.
>
> [2] R. Scheibler, et al, Pyroomacoustics: A Python package for audio room simulations and array processing algorithms, Proc. IEEE ICASSP, Calgary, CA, 2018.

---

> > ### Comment · Reviewer_RoFe · 2022-08-08
> > **Some concerns addressed**
> >
> > Thank the authors for the response, which addresses many of my concerns. The new comparions are very helpful. It makes sense to use a compact neural representation to encode the acoustic fields, which can be computationally expensive to simulate. However, I am still not very convinced that the dataset used in the paper is the best testbed here, because the main argument is that the neural network can render very *realistic and accurate* sound in real-time. And just looking at the qualitative results shown on the loudness map is not very convincing to show that the model works well on the points where there are no ground-truth. Having said that, the paper proposes some interesting ideas, and the proposed modules make sense and have inductive biases built into the network. I am raising my score and would be fine to see this paper accepted.

---

### Official Review · Reviewer_ddeR · 2022-07-11

**Rating:** 5
**Confidence:** 3
**Soundness:** 2 fair
**Presentation:** 1 poor
**Contribution:** 2 fair

**Summary:**

In this paper the authors address the problem of generating high fidelity impulse responses for acoustic scene rendering for AR/VR and teleconferencing type of applications which rely on faithful recreation of acoustic scenes. The basic idea revolves around decomposing RIR generation into two sets of attributes: i) attributes pertaining to source and receiver locations and ii) attributes pertaining to the room geometry. Features are separately extracted for these attributes using a feed-forward net. Extracted features are then concatenated to estimate binaural impulse responses using another feed-forward net. All the FF nets are trained end-to-end using waveform reconstruction loss and a multi-resolution spectro-temporal reconstruction loss which accounts for reconstruction of magnitude and phase spectrograms separately. The proposed method is compared to audio coding standards and relatively new deep learning based approach for RIR generation (Neural Acoustic Fields - NAF). Experimental results show that the proposed decomposition of attributes and network training is effective in generating high quality impulse responses, reduce storage requirements, fast and can be used to generate IRs in multiple rooms of different shapes.

**Questions:**

1. At a fundamental level, from the input-output specifications of the network it appears that the function that the network is trying to approximate is a one-to-many function as for a specified input - of emitter+receiver locations and bounce points, acoustic conditions can be changed by changing the reflection co-efficients of the walls. So are these networks trying to memorize a specific enclosure ? In this case it is not generalizable to other rooms with exact same geometry but with different reflection characteristics ?
2. Would it be possible to assess the generalization capabilities of the network in the true sense by generating RIRs of unseen environments ?
3. Was the point of comparing the proposed approach with AAC and Opus to say merely storing discrete IRs and interpolating will not help ?
4. Generically stating 40-60 bounce points would suffice is not making sense to me. Is this not dependent on room geometry in general and size in particular ?
5. in line 226, shouldn't \alpha_k bea function of i ?
6. In Fig 3, d_{b_{i}}^s is this a vector of distances or difference in position vectors. Dimesnions of these inputs should be clarified. Similarly for the gather modules.

**Limitations:**

A few limitations are raised in Questions section. Based on authors' response I will edit this section at a later stage.

**Strengths And Weaknesses:**

Strengths:
1. The idea of decomposing RIR attributes separately for room and source/receiver locations is very interesting. The idea of using Bounce points and the three modules to capture interaction of emitter with the room (bounce points), propagation from bounce points, and receiver with bounce points is also very interesting.

Weakness:
1. The paper is not very easy to read. I believe the presentation can be greatly simplified.
2. Goal of the paper is not crisply stated and is scattered throughout the paper.
3. Purpose of comparing with Audio coding standards is completely lost on the reader.
4. Abstract is not very informative.
5. Generalization experiments are a misnomer. Those experiments are mostly multi-condition training than generalization.

---

> ### Author Response · Authors · 2022-08-02
> **Response to Reviewer ddeR**
>
> ### Questions
> - Q1. Fundamentally, INRAS implicit neural representation is the parameterization of a continuous acoustic field using neural network parameters. The acoustic field is encoded within the neural network, providing a more compact representation. Since the Soundspace dataset fixed the reflection coefficient before simulating sounds, we didn’t consider such variations. \
> To answer this question, we tested whether INRAS can be conditioned on various reflection coefficients by creating a 3m x 5m room using the available open source python package (pyroomacoustics) [1] to simulate impulse responses using Image Source + Ray tracing method. We use spatial resolution of 0.5m to sample both impulse responses and bounce points. For the same room, we created 225 sets of impulse responses with various absorption and scattering coefficients on the walls. We use 90% sets of coefficients to train and 10% to test. For the coefficients seen in the training, we also hold 10% spatial points for testing. To incorporate the coefficient condition in our model, we adapt our model by adding a coefficient embedding using sinusoidal embedding similar to coordinate inputs. An addition operation is performed to add the coefficient embedding to the decomposed features. In this experiment, as the following table, INRAS model achieved testing t60 error of 3.83%, C50 error of 1.13dB and EDT error of 0.03 sec, which is quite close to the performance on the training set. These results show that INRAS can perform well even when absorption and scattering coefficients are unseen and are varying.
> | Settings                                          | T60 error | C50 error | EDT error |
> |---------------------------------------------------|-----------|-----------|-----------|
> | Seen Varying Reflection Coefficients (Training)   | 3.56      | 1.06      | 0.023     |
> | Unseen Varying Reflection  Coefficients (Testing) | 3.83      | 1.13      | 0.030     |
> - Q2. A complete generalization to an unseen environment without any condition is challenging for any type of implicit neural representation including INRAS. Here, we focused on modeling a compact representation for known scenes. However, as discussed above, we do assess several aspects of generalization of the pre-trained INRAS model such as variation of physical properties of the scene, slight geometrical variations, etc. \
> If the scene is completely unseen and significantly different than those that INRAS was pre-trained on, the same quality of the generated impulse responses as for the seen scenes cannot be guaranteed. \
> Indeed, when we tested such an unseen scenario, the evaluation showed a large T60 error between the generated IR and the ground truth. We will add this discussion to the limitation section.
> - Q3. See response to W3.
> - Q4. The number of bounce points is dependent on the room size and the sampling rate. We performed experiments with various sampling rates to evaluate which selection to use. For example, ‘Room_2’ has a perimeter of about 24 meters and the total number points of the mesh boundary is ~ 2400. Our experiments indicated that uniform sampling of the bounce points every 0.5m (sample rate:1/50, 48 bounce points) is optimal. Notably, the impulse response data has the same spatial resolution.
> Using a higher sampling rate doesn’t achieve better performance because noise and redundant features are induced as the sampling resolution is higher than the spatial resolution. We compare different sampling rates for bounce points for the mesh of ‘Room_2’ in the following table, which will be added to the supplementary information along with the above discussion.
> | Sampling Rate | T60 error | C50 error | EDT error |
> |---------------|-----------|-----------|-----------|
> | 1/100         | 1.47      | 0.53      | 0.019     |
> | 1/75          | 1.42      | 0.53      | 0.018     |
> | **1/50**      | **1.41**  | **0.51**  | **0.016** |
> | 1/25          | 1.41      | 0.59      | 0.019     |
> | 1/10          | 1.70      | 0.71      | 0.025     |
>
> - Q5. In line 226, the alpha_k should be alpha_{i,k}.
> - Q6. In Fig 3, d_{b_{i}}^s is the difference between position vectors.
>
> [1] R. Scheibler, E. Bezzam, I. Dokmanić, Pyroomacoustics: A Python package for audio room simulations and array processing algorithms, Proc. IEEE ICASSP, Calgary, CA, 2018.

---

> ### Author Response · Authors · 2022-08-02
> **Response to Reviewer ddeR**
>
> We thank the reviewer for a thoughtful review and valuable feedback. We address the questions and specify the intended revisions related to the presentation of our work below.
> ### Strengths And Weaknesses
> - W1. We will follow the reviewer’s recommendations to clarify the presentation of the methodology in the Methods section. We will also extensively proofread the manuscript for overall grammatical and typographical amendments.
> - W2. Indeed, the main goal of our work is to introduce a light-weight neural network to learn a continuous implicit function to represent the acoustic fields of scenes. For that purpose, we propose a novel approach to compress acoustic fields in terms of required memory and rapid rendering of impulse responses for interactive applications.
> - W3. We compare INRAS to audio coding standards since INRAS is effectively a novel encoding method to compress the acoustic field of a scene. We thereby estimate the effectiveness of INRAS versus common audio coding standards. The comparison shows that the storage requirement of Audio coding standards is much larger than of INRAS (180-350MB vs. 3MB). Moreover, an interpolation approach is necessary for audio coding standards to generate impulse responses at unseen locations. We examined how interpolation of the training set encoded by Audio coding standards (AAC, Opus) impacts the quality of the generated impulse responses for the test set and we find that the quality drops significantly. These experiments demonstrate that INRAS should be more beneficial as an encoding method for acoustic fields of scenes than Audio coding standards.
> - W4. We will revise the abstract to more effectively emphasize the following key points and contributions of our work:
>   - We introduce a light-weight neural network to learn a continuous implicit function to represent the acoustic fields of scenes.
>   - INRAS disentangles scene’s geometry features.
>   - INRAS includes three modules to generate independent features for the emitter, the geometry of the scene, and the listener. This leads to efficient reuse of scene-dependent features for arbitrary emitter-listener positions.
>   - INRAS supports effective multi-condition training for multiple scenes by adding only a few trainable parameters.
>   - INRAS outperforms existing approaches on all metrics of audio rendering, including the impulse response quality, inference speed, and storage requirements.
> - W5. We thank the reviewer for pointing out that the term ‘generalization’ used to describe the option of using a single unified network to represent multiple scenes is too broad. We agree that a more suitable term could be used to describe this option. When referring to this option in the camera-ready revision of the manuscript we will use the term ‘multi-condition training’ and we will also clarify the exact meaning of this term. In addition to multi-condition training, we also consider additional aspects of INRAS extensions for unseen scenarios, e.g. unseen reflection coefficients. We describe these in experiments below in response to Q1. We will include these in the revised version of supplementary information.

---

### Official Review · Reviewer_6t7L · 2022-07-12

**Rating:** 8
**Confidence:** 5
**Soundness:** 4 excellent
**Presentation:** 3 good
**Contribution:** 4 excellent

**Summary:**

Sound in a 3D scene depends on the emitter position and the listener position. The spatial acoustic information of the scene is important for having an immersive experience during scene modeling wherein how sounds from a particular location in a scene are perceived in another location in the scene. The authors designed INRAS: implicit neural representation for audio scenes as a way to model the spatial information. Given a geometry of the scene, emitter and listener positions, INRAS learns the Binaural impulse response which is combined with the source sound to render the spatial sound.  The INRAS system has two main components where the first component learns the audio scenes feature decomposition through the scatter, bounce, and gather module. Here the bounce point is any point in the surface of the geometry environment from which the sound can bounce from the emitter to the listener. The second component is the listener module that combines the decomposed feature representation from scenes by fusing the three components via concatenation of the features associated with each bounce point b to predict the Spatial Binaural impulse responses taking into account the head orientation and the location of the left/right ear for the listener.

The disentangling of the feature representation in the first stage into scatter, bounce, and gather modules allows the model to generalize to multiple scenes with few trainable parameters. The proposed INRAS method outperformed the existing approaches on all the metrics of audio rendering and has greatly improved the inference, speed, and storage requirements since the model require less than 3 Mb of storage capacity with almost a 4-fold improvement in inference speed. Additionally, the authors provide evidence on why the three stages are necessary such as the removal of the bounce module eliminates the static scene feature impairing performance.

**Questions:**

Authors have performed experimentation with INRAS where the optimal number of bounce points mentioned in the paper is 40 to 60 and it is unclear how the authors arrived at this conclusion. Why would the performance of the system drop with the addition of the bounce points? Wouldn’t a system with 100 bounce points extracted from the mesh boundary that covers the geometry of the environment be better?

How would the modeling be changed/affected when the listener is placed in rooms of different sizes beyond the layouts that are covered in the Soundscapes dataset? How does the current approach take into account the number of obstacles in the room is unclear?

**Limitations:**

The authors provide a limitation for their work where the boundary of the scene should be given as input to the INRAS system which I agree is a primary limitation. The above limitation is not a problem for scenes with 3D models where the boundary can be inferred if not given but the scenes without any geometry and recorded impulse response. The INRAS method would not work in these situations since the bounce points will be impossible to define in situations with unknown geometry.  The method is dependent on the availability of datasets such as SoundScapes which imply the need for INRAS to have a sufficient amount of training data to learn a reasonable acoustic representation of a given scene. This type of data collection is a problem in the case of real-world situations where a large number of impulse responses will be required for INRAS to work.

With regard to the negative societal impact, the authors highlight a concern where rendered spatial audio can be used to manipulate the original nonspatial sound and create a non-authentic impression of the audio requiring the need for an additional authentication step on the output of INRAS preventing the unethical or illegal use.

Another concern that I think authors need to consider is whether it is possible to deduce the environment in which the listener is seated where the malicious user may only require the need of spatial sound - the output of INRAS.

**Strengths And Weaknesses:**

The paper provides a good description of the related works and clarifies how the INRAS method is a light-weighted and efficient neural network module that can generate high fidelity spatial impulse response at the arbitrary emitter and listener positions.

Strengths:
The model is a lightweight and efficient neural network that produces spatial impulse responses at arbitrary emitter-listener positions. INRAS is capable of modeling continuous implicit function which maps the corresponding time-domain directional and binaural impulse responses of the sound field.

The training step leverages the fact that the geometry of the scene is static and thus it is possible to learn reusable scene-dependent feature representations which can be associated with the emitter and the listener. The design of the two-stage approach to first generate independent scene geometric features and then during the second stage fuse the features to generate a binaural impulse response is a novel and logical step in building such an acoustic representation.

The approach generalizes to multiple scenes as the emitter and listener is made aware of scene geometry via the computation of the relative distance to bounce points in the scatter and gather modules. Here the bounce module will provide a static scene-dependent feature representation. The co-ordinate space of multiple scenes is normalized and the bounce points are collected from the different scenes to achieve generalization as the emitter and listener can realize the scene they are a part of.

Evaluation metrics cover the three aspects such as impulse response quality, storage requirements, and inference speed and on all three metrics the approach outperforms the NAF method. The model has a significantly lower C50 error and has a third of the parameters with less than 3 Mb required to store the model and a 4 fold improvement in inference speed due to the smaller size. Additionally, the method outperforms all other relevant approaches when it comes to generalizing to multiple scenes leading to significant improvement in terms of SNR and PSNR for INRAS system.

The paper is well written and organized with no/few grammatical errors and the supplementary section provides necessary information complementing the main paper.

Weakness:
The approach is dependent on the data and availability of the training data for any given number of scenes and thus limited in terms of scaling the system to any scenario.

The choice of the bounce points is important as the authors show that when bounce points are chosen with missing some of the boundaries the performance reduces for the INRAS representation as the T60 error and the C50 error increase. This raises the question on what would happen to model performance in a situation where the complete boundaries of the 3D scene are not defined.

---

> ### Author Response · Authors · 2022-08-02
> **Response to Reviewer 6t7L**
>
> We thank the reviewer for a thoughtful review, valuable feedback and acknowledgement of our work. We provide point by point clarifications and answer questions below.
> ### Strengths and Weaknesses:
> - W1. Our approach is indeed dependent on the availability of training data. This is not an issue for virtual scenes since sound simulation can be performed to generate data prior to training. For real world scenes, there typically would be less data available since it cannot be easily generated. Plausible solutions that are currently being used are physical scanning of real world scenes and obtaining 3D meshes and then performing sound simulations to generate training data. The scenes in the Soundspace dataset are such real world scenes.
> ### Questions:
> - Q1. Bounce points: The number of bounce points is dependent on the room size and the sampling rate. We performed experiments with various sampling rates to evaluate which selection to use. For example, ‘Room_2’ has a perimeter of about 24 meters and the total number points of the mesh boundary is ~ 2400. Our experiments indicated that uniform sampling of the bounce points every 0.5m (sample rate:1/50, 48 bounce points) is optimal. Notably, the impulse response data has the same spatial resolution.
> Using a higher sampling rate doesn’t achieve better performance because noise and redundant features are induced as the sampling resolution is higher than the spatial resolution. We compare different sampling rates for bounce points for the mesh of ‘Room_2’ in the following table, which will be added to the supplementary information along with the above discussion.
> | Sampling Rate | T60 error | C50 error | EDT error |
> |---------------|-----------|-----------|-----------|
> | 1/100         | 1.47     |  0.53     |  0.019    |
> | 1/75          | 1.42      | 0.53      | 0.018     |
> | **1/50**      | **1.41**  | **0.51**  | **0.016** |
> | 1/25          | 1.41      | 0.59      | 0.019     |
> | 1/10          | 1.70      | 0.71      | 0.025     |
> - Q2. Modeling rooms of different sizes beyond the layout covered in the *Soundspace* dataset is straightforward. The only setting that is required is the distribution of the bounce points depending on the room geometry and size and uniform sampling can be used as we used for *Soundspace* rooms.
> - Q3. We uniformly sample bounce points on the boundaries of obstacles appearing in the scene and there could be any number of obstacles. In the *Soundspace* dataset, the only obstacles are walls that block sound propagation but INRAS could also deal with additional sound blocking objects such as furniture. The distribution of bounce points on the obstacles is according to uniform sampling rate as discussed above.

---

### Meta-Review · Area_Chair_sXAG · 2022-08-27

**Recommendation:** Accept
**Confidence:** Certain

**Metareview:**

This is a technically good paper, with some flaws. Parts of the paper are hard to read.   Several questions remain, e.g. how to determine the optimal number and location of bouncepoints, and how they depend on room layout and content.

The motivation behind some of the comparisons, e.g. to AACs is unclear.

Regardless, the overall paper presents a well-defined novel idea, and represent a significant contribution.  The authors have also addressed most of the reviewers' comments satisfactorily.

I am recommending that the paper be accepted.



**Award:**

No

---

### Decision · Program_Chairs · 2022-09-14

Accept